# Histone Deacetylase Inhibitors and Phenotypical Transformation of Cancer Cells

**DOI:** 10.3390/cancers11020148

**Published:** 2019-01-27

**Authors:** Anna Wawruszak, Joanna Kalafut, Estera Okon, Jakub Czapinski, Marta Halasa, Alicja Przybyszewska, Paulina Miziak, Karolina Okla, Adolfo Rivero-Muller, Andrzej Stepulak

**Affiliations:** 1Department of Biochemistry and Molecular Biology, Medical University of Lublin, Chodzki 1 St., 20-093 Lublin, Poland; joanna.kalafut@umlub.pl (J.K.); estera.okon@umlub.pl (E.O.); jakub.czapinski@umlub.pl (J.C.); martaahalasa@gmail.com (M.H.); alicja.przybyszewska@umlub.pl (A.P.); paulina.miziak@umlub.pl (P.M.); adolfo.rivero-muller@umlub.pl (A.R.-M.); andrzej.stepulak@umlub.pl (A.S.); 2Postgraduate School of Molecular Medicine, Medical University of Warsaw, Trojdena 2a St., 02-091 Warsaw, Poland; 3The First Department of Gynecologic Oncology and Gynecology, Medical University of Lublin, Staszica 16 St., 20-081 Lublin, Poland; karolina.okla@umlub.pl; 4Tumor Immunology Laboratory, Medical University of Lublin, Staszica 16 St., 20-081 Lublin, Poland; 5Faculty of Science and Engineering, Cell Biology, Abo Akademi University, Tykistokatu 6A, 20520 Turku, Finland

**Keywords:** cancer, HDI, HDAC, EMT, MET, cadherin, catenin, vimentin, migration, invasion

## Abstract

Histone deacetylase inhibitors (HDIs) are a group of potent epigenetic drugs which have been investigated for their therapeutic potential in various clinical disorders, including hematological malignancies and solid tumors. Currently, several HDIs are already in clinical use and many more are on clinical trials. HDIs have shown efficacy to inhibit initiation and progression of cancer cells. Nevertheless, both pro-invasive and anti-invasive activities of HDIs have been reported, questioning their impact in carcinogenesis. The aim of this review is to compile and discuss the most recent findings on the effect of HDIs on the epithelial-mesenchymal transition (EMT) process in human cancers. We have summarized the impact of HDIs on epithelial (E-cadherin, β-catenin) and mesenchymal (N-cadherin, vimentin) markers, EMT activators (*TWIST*, *SNAIL*, *SLUG*, *SMAD*, *ZEB*), as well as morphology, migration and invasion potential of cancer cells. We further discuss the use of HDIs as monotherapy or in combination with existing or novel anti-neoplastic drugs in relation to changes in EMT.

## 1. Introduction

Epithelial-mesenchymal transition (EMT) is a biological reversible process in which cells undergo multiple biochemical changes—lose their epithelial properties, including cell-cell adhesion and cell polarity, and acquire mesenchymal phenotype, including the ability to invade the extracellular matrix (ECM) and potentially migrate to the distant places. Induction of EMT includes reorganization of cytoskeleton proteins, activation of transcription factors and production of extracellular matrix-degrading enzymes [1,2]. Recent studies revealed the large role of epigenetic mechanisms including DNA methylation, chromatin rearrangement, histone modifications and non-coding RNAs in the initiation and progression of cancers [3]. Histone modifications play important roles in gene expression regulation via changes in chromatic structure and recruitment of epigenetic modulators, which also controls phenotypic transformation. Abnormal histone modification patterns are closely associated with numerous diseases including cancers, thus they are considered promising biomarkers [4].

Histone deacetylase inhibitors (HDIs) are effective anti-cancer agents which, in monotherapy and/or in combination with conventional chemotherapeutics, exhibit anti-neoplastic properties through cell-cycle arrest, inhibition of migration and invasion, induction of differentiation and apoptosis in many types of cancer cells [5,6,7,8]. Combinations of HDIs with e.g., thienotriazolodiazepine (JQ1), an inhibitor of bromodomain-containing acetylation reader proteins like bromodomain-containing protein 4 (BRD4), have shown efficacy in several cancer types, including urothelial carcinoma [9]. It has been reported that HDIs can reverse EMT, a process called mesenchymal-epithelial transition (MET), through, inter alia, unblocking of *E-cadherin* repression in solid cancers [10]. Thus, suggesting that HDIs have a therapeutic role in inhibition of EMT in cancer cells [11,12,13,14]. However, conflicting results have been also found, where HDIs induced EMT by reversing stem cell-like properties and enhanced metastasis [15]. In this review we discuss the impact of various HDIs on epithelial and mesenchymal markers, as well as on migration and invasion of cancer cells (Figure 1). The efficacy of HDIs has been demonstrated in both in vitro and animal models in monotherapy and/or in combination with existing or novel chemotherapeutics.

## 2. Histone Deacetylases (HDACs) and Histone Deacetylase Inhibitors (HDIs)

Epigenetic regulation of gene expression is largely modulated throughout chromatin and nucleosome remodeling, which involves histone post-translational modifications (PTMs) [16]. These PTMs result in dynamic shifts between transcriptionally active and suppressed states of chromatin [17]. Histone PTMs include methylation, phosphorylation, acetylation, sumoylation, ubiquitination and ADP-ribosylation [18]. Histone acetylation, one of the most extensively studied PTMs of histones, is regulated by the balance between histone deacetylases (HDACs) and histone acetyltransferases (HATs) (Figure 2A) [16]. HATs are enzymes that transfer an acetyl group from acetyl-CoA to ε-amino lysine residues located on N-terminus of histones [19]. In contrast, HDACs are responsible for removing the acetyl group from the acetylated lysine residues. This reversible reaction is crucial for chromatin structure stabilization and transcriptional regulation of gene expression [20,21]. Histone hypoacetylation by HATs leads to an open chromatin conformation, which is easily available for transcription factors, through abolition of the positively charged residues of histones and negatively charged DNA. HDACs promote transcriptional silencing through deacetylation and thus chromatin compression (Figure 2B) [22].

### 2.1. HDACs

The 18 HDACs in humans are classified into four classes according to their sequence homology with yeast proteins and cofactor dependency [23]. The class I shares common domains with yeast transcriptional regulator RPD3 and includes HDAC1, HDAC2, HDAC3 and HDAC8 and they are placed into the nuclear compartment. Class II of HDACs is shared into two subclasses (IIa and IIb) and is closely related with HDA1 in yeast [24]. The class IIa encompasses HDAC4, HDAC5, HDAC7, and HDAC9, whereas class IIb includes HDAC6 and HDAC10 [25,26]. Class IIa HDACs are inactive on acetylated substrates, thus differing from class I and IIb enzymes. It has been demonstrated that class IIa HDACs are very inefficient enzymes on standard substrates [27,28]. Class II HDACs (HDAC4, HDAC5, HDAC7 and HDAC9), compared to HDAC class I, possess limited enzymatic activity on their own. Instead, they appear to act as gene-specific transcriptional corepressors mainly as components of multiprotein complexes [29,30,31]. The class II HDACs migrate between cytoplasm and nucleus. The class III (sirtuins) includes seven members (SIRT1-SIRT7) and they share common domains with yeast silent information regulator 2 (SIR2) [27]. Class IV contains only one member - HDAC11 [32]. Catalytical activity of class I, II, IV is strongly associated with presence of zinc ion in their active site. In contrast, class III requires nicotinamide adenine dinucleotide (NAD) as a cofactor during their catalytical reaction [25,26].

Imbalances between the activities of HDACs and HATs are associated with a plethora of diseases [33,34,35,36]. The epigenetic aberrations of gene expression caused by increased activity of HDACs play a pivotal role in cancer development and progression [37]. Given the fact that the activity of HDACs is dysregulated in many types of cancers [38,39,40], HDACs have been considered as therapeutic targets for the treatment of neoplasms, indeed HDIs have become promising anti-cancer agents [41,42].

According to the Human Protein Atlas class I HDACs are expressed in variety types of tumors (Figure 3) [43,44,45,46]. The data is presented by the percentage (%) of analyzed tumors with HDACs expression at high or medium level. HDAC2 was found expressed in 100% of multiple tumors. In the case of renal cancer HDAC2 is expressed 100% of cases, while other members do not exceed 50% (Figure 3) [44]. HDAC1 expression is comparatively high with exception of renal cancer and glioma [43]. The amount of patients with HDAC3 expression in ovarian cancer is significantly lower (20%) comparing to the other members of class I—all others with almost 100% incidence [45]. The last member of class I, HDAC8 is absent in colorectal, testis and breast cancer. Additionally, the score for patients with HDAC8 expression in liver cancer is notably lower (9%) compared to other members—HDAC1: 90%, HDAC2: 75% and HDAC3: 50% (Figure 3) [43,44,45,46].

Class II HDACs are more varied than class I in terms of incidence in different cancers. High incidence of class IIa HDACs (HDAC4, HDAC5 and HDAC9) [47,48,49], is associated with colorectal and breast cancers. Interestingly, very low level of patients with HDAC5 (9%) and HDAC9 (8%) expression is observed in renal cancer (Figure 4) [48,49]. The last member of IIa class—HDAC7 is associated mainly with lung cancer. Inhibition of HDAC7 results in restraining of lung cancer development [50]. HDAC10 (IIb class) is expressed in virtually all patients’ tumors (100%) in every single analyzed type of cancer [51]. In contrast, the number of tumors with HDAC6 expression is very diverse in different types of cancer [52].

SIRT3, SIRT5, SIRT6 and SIRT7 are expressed in large part of tumors [53,54,55,56]. Of note is SIRT2, which is only expressed with gliomas among all analyzed types of cancer (Figure 5) [57].

There is no available data in Human Protein Atlas regarding the expression of HDAC11 in different tumors. Yet, depletion of HDAC11 has an impact on cancer cells, including breast, ovarian, colon and prostate cells. HDAC11 is associated with apoptosis induction and inhibition of cell metabolic activity. Conversely, depletion of HDAC11 does not affect colon HCT-116 and prostate PC-3 cells [58].

According to the Human Protein Atlas high or medium expression of HDAC1, HDAC2, HDAC9, HDAC10, SIRT3, SIRT5, SIRT6, SIRT7 is present in 100% of patients with breast cancer (Figure 6A); while in carcinoid tumors is HDAC1, HDAC4, HDAC9, SIRT7 (Figure 6B). In cervical cancer HDAC1, HDAC10, SIRT6, SIRT7 are always present (Figure 6C); whist in colorectal cancer is HDAC10, SIRT6, SIRT7 (Figure 6D). Endometrial cancer only expresses in 100% of cases HDAC10 and SIRT3 (Figure 6E). 100% penetrance of HDAC2, HDAC4, HDAC10 and SIRT3 in glioma (Figure 7A); HDAC2, HDAC9, HDAC10, SIRT3, SIRT5, SIRT6, SIRT7 in head and neck cancer (Figure 7B); HDAC10, SIRT3, SIRT5 in liver cancer (Figure 7C); HDAC1, HDAC10, SIRT6, SIRT7 in lung cancers (Figure 7D); HDAC1, HDAC2, HDAC4 in lymphoma (Figure 7E); HDAC1, HDAC2, HDAC10, SIRT3, SIRT5, SIRT6 in melanoma (Figure 8A); HDAC9, HDAC10, SIRT3, SIRT6, SIRT7 in ovarian cancer (Figure 8B); while HDAC1, HDAC10, SIRT3, SIRT6 in pancreatic cancer (Figure 8C).

The same goes for HDAC1, HDAC2, HDAC10, SIRT3, SIRT6 in prostate cancer (Figure 8D); HDAC2, HDAC10 and SIRT6 in renal (Figure 8E) and skin cancers (Figure 9A); HDAC10, SIRT3, SIRT6, SIRT7 in stomach cancer (Figure 9B); HDAC2, HDAC5, HDAC10, SIRT3, SIRT6 in testicular cancer (Figure 9C); HDAC1, HDAC4, HDAC10, SIRT 3, SIRT5, SIRT6, SIRT7 in thyroid cancer (Figure 9D) and HDAC1, HDAC2, HDAC10, SIRT3, SIRT6, SIRT7 in urothelial cancer (UC) (Figure 9E) [43,44,45,46,47,48,49,50,51,52,53,54,55,56,57,58,59,60]. In urothelial cancer, not only up-regulation of HDAC2 and HDAC8, but also down-regulation of HDAC4, HDAC5 and HDAC7 mRNA are common findings. Selective targeting of HDAC2, HDAC8 and other HDACs dysregulated in UC result in a more consistent treatment response requires further research [61]. However, neither specific pharmacological inhibitors nor siRNA-mediated knockdown of HDAC8 reduced the viability of urothelial cancer cells (UCC), suggesting HDAC8 in not a good target for UC therapy [62,63].

Histone targets for HDACs are: H3K9Ac (acetylation in lysine 9 of histone 3), H3K18Ac, H4K5Ac, H4K8Ac, H4K12Ac and H4K16Ac in lung cancer, H3Ac, H4Ac and H3K18Ac in prostate cancer, H3K18Ac, H4K12Ac and H4K16Ac in breast cancer [64].

HDACs also deacetylate non-histone proteins [65,66]. Acetylation of non-histone proteins is a part of key cellular process in physiology and diseases, and links with signal transduction, gene transcription, metabolism, DNA damage repair, cell division, autophagy and protein folding. Acetylation affects the function of proteins through various mechanisms, including regulation of protein stability, enzymatic activity and crosstalk with other post-translational modifications [66]. One of the non-histone target of HAT acetylation is the tumor suppressor p53. Acetylation of p53 by p300/CBP (CREB-binding protein) activates its sequence-specific DNA binding activity and increases activation of its target genes. Deacetylation of p53 by SIRT1 decreases the ability of p53 to activate the cell cycle inhibitor p21, which takes part in DNA repair [66,67]. YY1 is sequence-specific DNA-binding transcription factor involved in repressing and activating a diverse number of promoters. YY1 interacts with HATs (CBP and p300) and with most HDACs class I (HDAC1, 2 and 3) [67,68]. Moreover, acetylation regulates the DNA binding activity of high mobility group (HMG) proteins. In metastatic human colon adenocarcinoma cells HMGA-1 proteins are more highly acetylated in comparison to the non-metastatic precursors [69,70].

Nuclear receptors (NRs) are the other class of transcription factors modulated by acetylation and deacetylation. CBP/p300 and TIP60 acetylate the androgen receptor (AR). Hormone-dependent activation of AR requires acetylation of lysines 630, 632 and 633. Deacetylation of AR by HDAC1 represses the function of AR [71]. The estrogen receptor (ER) is also acetylated by p300 but at lysines 299, 302 and 303 [67]. Another non-histone protein GATA-1 is acetylated by p300. GATA-1 is an important transcription factor in hematopoiesis and terminal differentiation of erythrocytes and megakaryocytes [72]. Erythroid Krüppel-like factor (EKLF) is a red cell-specific transcriptional activator. EKLF is acetylated and interacts with p300, CBP and P/CAF [73]. p300/CBP acetylates EKLF at lysine residues 288 and 302 located in the transactivation domain and zinc finger domain, respectively [67]. The myogenic protein (MyoD) requires CBP/p300 and PCAF acetylation to transactivate muscle-specific promoters [67]. The proliferation promoting members of the E2F family (E2F1, 2 and 3) also are acetylated by p300, CBP and PCAF, the latter acetylates E2F1 with the highest efficiency [67,73]. Acetylation and deacetylation also dynamically regulate the activity of NF-κβ (nuclear factor kappa-light-chain-enhancer of activated β cells). NF-κβ is a protein complex that controls cell survival, transcription of DNA and cytokine production. The nuclear function of the NF-κβ transcription factor is regulated through acetylation of its RelA subunit by p300/CBP at the lysines 218, 221, 310. Acetylation of lysine 221 in RelA subunit enhances DNA binding and impairs assembly with Iκβα. While, acetylation at lysine 310 is needed for full transcriptional activity of RelA in the absence of effects on DNA binding and Iκβα assembly. Site-specific acetylation of RelA diversely regulates activities of NF-κβ the transcription factor complex [74].

Acetylation regulates also activity of the molecular chaperone Hsp90. Hsp90 has important role in maturation of many proteins, including the ligand-inducible transcription factor glucocorticoid receptor (GR). Specifically HDAC6 seems to be a regulator of Hsp90 acetylation [67,75]. Moreover, hypoxia-inducible factor 1 (HIF-1) can be acetylated by ARD1 protein acetyltransferase. HIF-1 plays a main role in cellular adaptation to changes in oxygen availability. ARD1-mediated acetylation strengthen interaction of HIF-1α with pVHL (the von Hippel-Lindau protein) and HIF-1α ubiquitination, suggesting that the acetylation of HIF-1α by ARD1 is critical to proteasomal degradation [76]. Transforming growth factor beta (TGFβ) regulates multiple cellular processes via activation of Smad signaling pathways. p300 acetylates Smad7 on two lysine residues. These lysine residues are critical for Smurf-mediated ubiquitination of Smad7. Moreover, acetylation protects Smad7 from TGFβ-induced degradation [77]. It has been also demonstrated that p300/CBP acetylates mastermind-like transcriptional coactivator-1 (Maml1), a Notch transcriptional co-factor, and thus regulates the strength of Notch-downstream signaling [78]. On the other hand, Notch signaling induces SIRT2 expression, which deacetylates and activates ALDH1A1 (aldehyde dehydrogenease), a marker commonly used to determine stem cells, particularly in breast cancer [79].

### 2.2. HDIs

HDIs are divided into four basic structural classes: short chain fatty acids (e.g., valproic acid (VPA), sodium butyrate (NaB), phenylbutyrate (PBA)), hydroxamic acids (e.g., vorinostat (SAHA), trichostatin A (TSA), panobinostat (LBH-589), belinostat (PXD-101), resminostat (4SC-201)), cyclic peptides (e.g., romidepsin (FK228), apicidin (CAS183506-66-3)), benzamides (e.g., entinostat (MS-275), mocetinostat (MGCD103), domatinostat (4SC-202)) [80,81]. HDIs differ significantly in their specificity for HDACs (Table 1). Most HDIs belonging to benzamide analogs (MS-275, MGCD0103, 4SC-202) and cyclic peptides (FK228, CAS183506-66-3) groups inhibit HDAC class I members only, while the majority of HDIs which are short-chain fatty acid can inhibit HDAC classes I and II [82,83]. Some of hydroxamic acid-derived compounds (SAHA, 4SC-201, PXD-101) are HDAC pan-inhibitors (Table 1). Pan-inhibitors characterize the lowest specificity, therefore they can inhibit various HDACs belonging to different classes [82,83].

In the past decade, many HDIs have been found to possess powerful anti-cancer activity, including induction of apoptosis [92,93], growth arrest and differentiation [94], suppression of EMT, cell migration and invasion [13], as well as inhibition of angiogenesis [95], both in vitro and in vivo [42]. Additionally, HDI-induced suppression of tumor growth and apoptosis of neoplastic cells take place without noticeable effects in normal cells [5]. Currently, four HDIs—vorinostat, romidepsin (antibiotic) [96], belinostat and panobinostat—have been approved by the Food and Drug Administration (FDA) for the treatment of cutaneous and peripheral T-cell lymphoma and multiple myeloma [18]. Several HDIs are in various phases of clinical trials, either as monotherapy and in combination with existing or novel anti-cancer drugs [18]. The molecular mechanisms for the anti-cancer activity of HDIs have not been fully resolved, partly as their effects are cell type-, dose- and time-dependent. It is worth mentioning that HDIs do not only affect histone–DNA complexes, but also the acetylation status of non-histone proteins (e.g., STAT3, p53 transcription factors) [17,18,97].

## 3. Epithelial-Mesenchymal Transition (EMT)

EMT is an essential physiological process during embryogenesis, histogenesis, organogenesis and wound healing. Yet, it can be also exploited during pathological processes such as fibrosis or tumor progression [98,99]. EMT is a reversible cellular process where epithelial cells acquire a mesenchymal-phenotype. Epithelial cells are connected by intercellular junctions such as: desmosomes, tight junctions (TJ) and adherent junctions (AJ), in contrast to mesenchymal cells, which do not cling to each other [100,101]. The consequence of EMT is disappearance of adhesion between epithelial cells through loss of junctions structures and apical-basal polarization. The new-formed mesenchymal cells acquire high migratory capabilities and invasive properties [102]. EMT is strongly associated with cancer metastasis as well as with presence of the circulating tumor cells (CTC). Moreover, EMT induces chemo- and radio-therapy resistance in many kinds of tumors [102,103]. Throughout EMT, the cancer cells endure frequently molecular events, for instance, a decrease of the level of epithelial markers (E-cadherin, cytokeratins) and an increase of the level of mesenchymal markers (N-cadherin, vimentin) (Figure 3). Expression of EMT markers in primary tumors has been linked with cancer progression and poor medical prognosis [104,105].

EMT is induced by growth factors including: transforming growth factor, hepatocyte growth factor, epithelial growth factor, fibroblast growth factor and insulin growth factor. All of them indirectly modify EMT transcription factors [106] (EMT-TFs), including: *SNAIL* and *ZEB1/ZEB2* families, as well as *TWIST1/TWIST2*. Vertebrates have 3 *SNAIL* family members: *SNAIL1*, *SNAIL2* and *SNAIL3*. These TFs have a highly conserved *C*-terminal organization, able to recognize and bind to the *E-cadherin* promoter. Moreover, the *N*-terminal domain of *SNAIL* (SNAG) interacts with transcriptional co-repressors, including Sin3A/HDAC1/2 complex and polycomb complex 2. Hence, the activation of *SNAIL/SLUG* promotes *E-cadherin* gene (*CDH1*) downregulation and contributes to an increase of cell migration and invasion [106,107]. The *ZEB* family of TFs downregulates *CDH1* expression and upregulates mesenchymal markers such as *N-cadherin* gene (*CDH2*), *vimentin* and *fibronectin*. *ZEB* members are also responsible for increase of cell migration and invasion [108]. *TWIST1* is able to simultaneously upregulate *CDH1* and downregulate *CDH2* expression. Post-transcriptional gene expression is regulated by small non-coding RNAs, such as: miRNA-200 and miRNA-34. Where epithelial cells express miRNA-200 and miRNA-34 whilst mesenchymal cells do not [109].

The balance between EMT and MET processes regulates cell plasticity [110]. However, nowadays an intermediate stage between fully-epithelial and fully-mesenchymal states has been recognized—hybrid E/M state. The identification of EMT/MET or hybrid E/M states is difficult to observe because these processes run smoothly and interchangeably [110] (Figure 10). Cancer cells with hybrid E/M phenotype have cell-cell adhesion properties as well as migration abilities, simultaneously [109]. Recent data suggest that cells with E/M hybrid phenotypes show stronger metastatic properties as well as survival in circulation [111,112]. Hybrid E/M cells are similar or more resistant to drug-treatments in comparison to fully EMT cells [111].

## 4. EMT and Cancers

EMT is the result of a series of epigenetic changes including chromatin remodeling and histone modifications. Acetylation and metylation of histones play an important role in tumor progression [41]. For this very reason, HDIs are considered as modifiers of EMT-related factors expression, although this effect is cancer-type dependent [113]. Hereby we will analyse the available data on HDIs in different tumor types.

### 4.1. Lung Cancer

HDIs have been investigated for their roles as inhibitors of migratory potential. Indeed, TSA inhibits migration of irradiated human epithelial A549 lung cancer cells through decreasing of SNAIL and ZEB expression. The expression of E-cadherin and N-cadherin in irradiated-cells treated with TSA are inverted as compared to radiation-only pretreatment. Radiation-TSA treatment also resulted in upregulation of ZO-1 and β-catenin (epithelial markers), compared with alone-radiation pretreatment [114]. Moreover, it has been shown that the inhibitory effect for TGF-β1-induced EMT in irradiated A549 cells pretreated with TSA is connected with inhibiting of SNAIL and SLUG activity [115]. TSA supported with silibinin, a natural flavanone compound from silymarin, significantly increases E-cadherin level by downregulation of ZEB1, while silibinin alone is not able to silence E-cadherin expression in non-small cell lung cancer (NSCLC H1299 cells). Interestingly, the level of E-cadherin after 48 h of TSA+silibinin treatment was significantly restored, compared with the level of E-cadherin after 48 h of TSA-alone [116]. VPA was able to partially inhibit EMT in A549 cells, through decreasing histone deacetylation level. Additionally, the cellular spindle-shape effect, which is characteristic for mesenchymal cells, induced by TGF-β is reduced after VPA treatment. Although there is no direct interaction between VPA and TGF-β1 [117].

At odds with other HDIs, SAHA-treated A549 cells responded by decreasing of E-cadherin expression and increasing of vimentin expression, with the acquisition of a mesenchymal phenotype. The E-cadherin downregulation is inversely correlated to SLUG expression [118]. Nevertheless, SAHA, as well as panobinostat, induce upregulation of GAS5-AS1 expression in a dose-dependent manner in NSCLC cells, which is connected with inhibiting migration of NSCLC cells [119] (Table 2).

### 4.2. Hepatocellular Carcinoma

TSA, VPA, SAHA and MS-275 have a strong positive influence on EMT, through decreasing E-cadherin expression and increasing N-cadherin expression in HepG2 cells. In turn, mesenchymal markers such as vimentin, TWIST and SNAIL become more abundant [120]. In the same vein, SAHA and sodium butyrate (NaB) have been investigated as suppressors for cells proliferation in dose-dependent manner. Both of them significantly increase N-cadherin, vimentin, fibronectin and SNAIL expression in HepG2 cells. SNAIL upregulation is connected with phosphorylation of SMAD2/3 by these HDI. Additionally, SAHA and NaB are able to promote SNAIL and vimentin expression in xenografs [121]. Yet, panobinostat (LBH589) elevates E-cadherin expression in HCC-LM3 and HepG2 cells while decreases N-cadherin, vimentin and TWIST1 simultaneously [122]. Likewise, reminostat acts as an upregulator of E-cadherin expression and down-regulator of vimentin, TWIST1 and SNAIL in HLE cells [123] (Table 2).

### 4.3. Cholangiocarcinoma

VPA or TSA increased both E-cadherin and vimentin expression but inhibited invasion and migration of HuCC-T1 cholangiocarcinoma cells. Additionally, HuCC-T1 cells co-treated with gemcitabine and VPA or TSA showed higher E-cadherin, vimentin and ZO-1 levels as well as decreased migration and invasion. 

Moreover, HuCC-T1 cells altered from spindle (mesenchymal phenotype) to rectangular (epithelial phenotype) shape after gemcitabine together with VPA or TSA treatments [124] (Table 2).

### 4.4. Pancreatic Cancer

HDAC inhibition by domatinostat (4SC-202) results in downregulation of E-cadherin with the concomitant upregulation of N-cadherin in Panc-1 cells, but unexpectedly the downregulation of other mesenchymal markers such as ZEB1, SNAIL and vimentin, or TGF-β-induced SMAD2 phosphorylation. Likewise, it induces ZEB1 and SNAIL1 downregulation and CD24 upregulation in L3.6 and PxPC3 cells [125]. Conversely, (3R)-2-(biphenyl-4-ylsulfonyl)-1,2,3,4-tetrahydroisoquinoline-3-carboxylic acid (BSI) increases E-cadherin expression while decreases N-cadherin and SNAIL expression after 24h in Panc-1 cells. Interestingly, the level of E-cadherin remains unchanged, although the level of N-cadherin and SNAIL was decreased in BxPC-3 cells after BSI treatment. Moreover, BSI is strongly associated with partial inhibition of invasion and migration in Panc-1 cells after 24 h. BSI reduces tumor spheres formation in BxPC-3 cells, while in Panc-1 cells spheres formation is unchanged but their size was significantly decreased [126]. It has been found that another interesting agent, mocetinostat inhibits ZEB1 expression and increases E-cadherin and miR-203 upregulation in Panc-1 cells as well as in hPaca1-derived tumor cells. Paradoxically Panc-1-tumor xenografts grew bigger by mocetinostat treatment while the combination with gemcitabine resulted in a synergetic effect in tumor growth inhibition [127]. SAHA, on the other hand, inhibits proliferation in pancreatic CSCs. SAHA is able to increase miR-34a expression in pancreatic CSCs as well as in ASPC-1 and able to increase miR-34a expression in pancreatic CSCs as well as in ASPC-1 and MiaPaCa-2 cell lines. SAHA induces E-cadherin overexpression and N-cadherin downregulation in pancreatic CSCs, simultaneously. Moreover, SAHA, as well as resveratrol, significantly downregulates ZEB1, SNAIL and SLUG expression in pancreatic CSCs. Additionally, resveratrol inhibited the invasion and migration of pancreatic CSCs. Resveratrol was able to inhibit the growth of pancreatic cancer in KrasG12D mice [12,144] (Table 2).

### 4.5. Colorectal Cancer

TSA has been studied for its effects on SW480 colorectal cancer cells. TSA decreased the expression of *SLUG*, leading to the reversal of the EMT process and attenuation of invasion and migration of SW480 cells. It has been suggested that TSA causes EMT reversion by increasing of E-cadherin and decreasing of vimentin expression [128]. Au contraire, treatment with VPA significantly stimulates migration and invasion in vitro, argubly by activation of EMT in HCT116 and SW480 human colorectal cancer cell lines, resulting in downregulating the epithelial markers: E-cadherin and ZO-1 and upregulating the mesenchymal markers: N-cadherin and fibronectin in both HCT116 and SW480 cells as well as upregulating the vimentin only in HCT116 cells. In line with this, VPA significantly promotes the expression of *SNAIL* via Akt/GSK-3b signal pathway. Suppression of *SNAIL* significantly reduced E-cadherin and increase of vimentin or fibronectin expression in both HCT116 and SW480 cells [128]. In fact, other HDIs also block EMT or induce MET, such as compound-11, who has also been found to induce MET in HCT116 and HT29 colorectal cancer cells, as well as in the HCT116 xenograft model. It has been observed that compound-11 induced downregulation of N-cadherin, vimentin and p-FAK (invasive marker), while E-cadherin was increased, through downregulation of Akt, which seems to be crucial for EMT in colorectal cancer cells [129]. Nevertheless, the oppsite has also been observed using TSA and VPA individually or in combination with TGF-β1 in four colon carcinoma cell lines including: SI cells (DLD1 and HCT116) and MSS cells (HT29 and SW480). The results revealed that the morphological changes were similar pursuing TSA or VPA with or without TGF-β1 co-treatment. CRC cell lines were altered to spindle-like morphology. Subsequent analyses showed a decrease in E-cadherin expression with TSA or VPA treatments in HCT116, DLD1 and SW480 cells. Vimentin was increased by treatment with the HDIs together with TGF-β1 in the four carcinoma cell lines. Consistently, TSA or VPA induced increased cell migration and invasion abilities. All together, treatment by TSA or VPA in combination with TGF-β1 seem to intensify EMT and migration in colon carcinoma cells. Moreover, in the MSS cells (HT29 and SW480) the EMT process was enhanced by TGF-β1 and was much more intense than in the MSI cells (DLD1 and HCT116) [15] (Table 2).

### 4.6. Renal Cancer

VPA or MS-275 treatment resulted in cell morphology alternation and a reduction in migration of in Renca cells as compared to untreated Renca cells. At the molecular level, *TWIST1* was upregulated and *TWIST2* was downregulated after MS-275 treatment in time-dependent manner. Moreover, *SNAIL2* expression was increased, while *SNAIL1* expression was unchanged after 48 h of 5 μL MS-275 treatment. Additionally, *ZEB2* was significantly increased in dose-dependent manner, while *ZEB1* remained unchanged after 48h of MS-275 treatment. VPA and MS-275 hardly decrease β-catenin expression. Moreover, both of them upregulated E-cadherin levels after 48h. Interestingly, VPA significantly increased the growing rate of Renca cells resulting in phenotypical changes in cell morphology. Untreated Renca cells had a cobblestone-like morphology. HDIs treatment altered their morphology to a scattered pattern, with interspaces between cells. These cells displayed a star-shaped cell body resembling EMT-associated growth [130]. MS-275 as well as TSA significantly increased E-cadherin expression in TGF-β1-treated HK2 cells. On the other hand, neither of them had no influence on N-cadherin expression in the same cells [131]. Moreover, TSA increases E-cadherin expression without any effect on *SMAD2* and *SMAD3* phosphorylation in RPTEC cells [145]. PCI34051 resulted in no changes on N-cadherin and E-cadherin expression in HK2 cells. Finally, LMK235 was able to restore E-cadherin expression in HK2 cells, which is downregulated by TGF-β1 [131] (Table 2).

### 4.7. Urothelial Carcinoma

The capacity of human urothelial cancer cell lines (UCCs) to form tumors after implantation on to the chicken chorioallantoic membrane (CAM) was examined. Both, RT-112 (epithelial-like) and T-24 (mesenchymal-like) urothelial cells generated tumors in the CAM model. RT-112 and T-24 cells in cell culture or as CAM tumors were treated with cisplatin alone or in combination with romidepsin or SAHA. Expression of E-cadherin (epithelial marker) and vimentin (mesenchymal marker) in untreated cells was similar in 2D cultures and CAM tumors. Cisplatin with HDIs reduced growth and weight of CAM tumors in a dose-dependent manner. HDIs treatment acted less efficiently in 2D cultures than in CAM model. Tumor size and weight were higher for RT-112 than T-24. Moreover, RT-112 tumors were more vascularized than T-24 tumors. RT-112 and T-24 CAM tumors were treated with IC_25_ and IC_50_ of cisplatin (CDDP) for 72 hours. Both weight and size of cisplatin-treated tumors were significantly reduced, especially in RT-112. Ki-67 mRNA expression in RT-112 cells was upregulated both in 2D cultures and CAM tumors after SAHA treatment. Downergulation of Ki-67 mRNA expression was observed in T-24 2D cultures treated with romidepsin or SAHA, but it was increased in HDIs-treated CAM tumors (Table 2) [132].

### 4.8. Prostate Cancer

AR-42 inhibited migration and invasion of Ace-1 cells caused apoptosis and decreased PCa cells bone metastasis. Moreover, AR-42 decreased E-cadherin, N-cadherin, *TWIST*, *MYOF*, and osteomimicry genes expression as well as anoikis (apoptosis induced by lack of correct cell/ECM attachment) resistance, while it increased *SNAIL*, *PTEN*, *FAK* and *ZEB1* transcription factors expression in Ace-1 cells. In addition, AR-42 downregulated the PCa metastasis to bone in nude mice. In addition, there has been observed an alteration of the spindle-like morphology to irregular shape of PCa cells, in both in vitro and in vivo conditions after AR42 treatment [133]. Furthermore, treatment with another HDI-SAHA-repressed EMT in LNCaP prostate cancer. It has been reported that SAHA downregulates *FOXA1* expression. *FOXA1* inhibits EMT in prostate cancer by decreased expression of *SLUG* transcription factor and repression of the neuroendocrine (NE) differentiation markers. SAHA also decreases *NKX1* and *PSA*, which is another transcription factor and antigen, respectively. SAHA, like other pan-HDAC inhibitors (inter alia TSA), induces EMT by elevated protein levels such as *SLUG*, *ZEB1* and vimentin. The same results were obtained using TSA treatment in LNCaP cells, however treatment with RGFP966, with the same panel of EMT markers was inefficient both in cell migration or invasion. SAHA and TSA induced cell migration, while RGFP966 was innocuous [134]. Treatment with LBH589 suppresses HMGA2 expression, decreases epithelial-mesenchymal plasticity in vitro and drastically decreases tumor growth and metastasis in vivo. Notably, in mice treated with LBH589 in combination with orchiectomy, there was an increase of p53 and androgen receptor (AR) acetylation, which in turn prevents the development of mCRPC and considerably extends life after castration [146]. In addition, TSA reverts EMT by a time-dependent upregulation of E-cadherin and downregulation of vimentin in PC3 prostate cancer cells. Moreover, TSA it has supressed *SLUG* expression which consequently prompted MET, as well as decreased cell invasion and migration abilities [13]. Likewise, VPA inhibited EMT by upregulation of the expression of E-cadherin, and concomitant suppression of the migration and invasion of prostate cancer cells [11] (Table 2).

### 4.9. Breast Cancer

SAHA inhibits EMT and chemoresistance induced by TGF-β1 in MzChA-1 and TFK-1 breast cancer cells. In both of these cell lines, TGF-β1 caused morphological changes from valvate-like to spindle-like shapes, as well as downregulation of E-cadherin and upregulation of N-cadherin, vimentin and *SNAIL* expression that the mechanism of SAHA’s effect seems to be the inhibition of *p-SMAD2*, *p-SMAD3* and *SMAD4* nuclear translocation induced by TGF-β1 in MzChA-1 cells, as well as the attenuation of the binding affinity of *SMAD4* to the E-cadherin-related *TWIST*, *SNAIL*, *SLUG*, *ZEB1* and *ZEB2* transcription factors [135]. Yet, SAHA can promote migration and EMT via HDAC8/FOXA1 signals in MDA-MB-231 and BT-549 breast cancer cells. SAHA significantly downregulated the expression of E-cadherin and upregulated the mesenchymal markers: N-cadherin, vimentin and fibronectin. However, SAHA had no effect on the nuclear translocation or expression of *SNAIL*, *SLUG*, *TWIST* and *ZEB* [136]. MDA-MB-231, BT-549 and MCF-7 breast cancer cells were incubated with LBH589, another HDI, and examined for changes in cell morphology, migration and invasion in vitro. LBH589 reversed EMT, measured by the altered morphology and gene expression of triple negative breast cancer (TNBC). E-cadherin expression was significantly upregulated by LBH589 treatment in the two TNBC lines (MDA-MB-231 and BT-549), while no change was observed in the ER-positive (MCF-7) cells. Additionally, expression of *ZEB1* and *ZEB2* were significantly inhibited upon LBH589 treatment in both the MDA-MB-231 and BT-549 TNBC cell lines, while no changes were detected in the MCF-7 cells. The above-described alterations in EMT gene expression correlated with diminished cell migration and invasion in TNBC cells in vitro, as well as meaningful inhibition of TNBC cell metastasis to lung and brain in a xenograft model [138]. Treatment of MDA-MB-231 and Hs578T cells with entinostat (ENT) caused upregulation of *CDH1* and downregulation of *CDH2* and *VIM* mRNA expression. Moreover, chromatin immunoprecipitation (ChIP) assay revealed that the treatment of MDA-MB-231 and Hs578T cells with ENT increased the reduced the association of *SNAIL* and *TWIST* to the *CDH1* promoter and downregulated both the *SNAIL* and *TWIST* expression which resulted in higher E-cadherin expression. Moreover, ENT inhibited migration of MDA-MB-231 and Hs578T breast cancer cells and induced MET [14]. The HDAC1 and HDAC3 inhibitor—MS-27—sensitized tumor necrosis factor-related apoptosis-inducing ligand (TRAIL)-resistant breast cancer MDA-MB-468 cells, inhibited angiogenesis and metastasis, and reversed EMT in vivo in xenografted BALB/c nude mice. MS-275 upregulated the expression of E-cadherin and downregulated the expression of N-cadherin, as well as *ZEB1*, *SNAIL* and *SLUG* transcription factors in tumor tissues. Treatment of mice with TRAIL alone had no effect on the expression of these markers. Co-treatment of MDA-MB-468 cells with MS-275 and TRAIL had similar effects to those of MS-275 [139]. Yet, it is in breast cancer where contradicting data exist. For example, SUM159 and MDA-231 cells treated with VPA or SAHA become more stem-like by dedifferentiation. These dedifferentiated cells have a higher migration potential and are more resistant to taxol. HDIs-treated cells presented upregulation of several mesenchymal markers such as vimentin, N-cadherin, fibronectin and tenascin-C while epithelial marker E-cadherin was not detected. Yet, several other mesenchymal markers such as *SNAIL* (after SAHA treatment), *FOXC2* and *ZEB1* (after SAHA and VPA treatment) were downregulated. HDACs inhibition resulted in the activation of the Wnt/β-catenin signaling, which seems to be responsible for these phenotypical changes [137] (Table 2).

### 4.10. Ovarian Cancer

Effects of TSA alone or in combination with cisplatin were investigated in SKOV3 cell line in vitro. SKOV3 cells showed downregulation of both E-cadherin and N-cadherin with exposure to TSA alone or in combination with cisplatin. Moreover, mouse xenografts were used to assess the anti-cancer activity of sequential cisplatin followed by TSA treatment. Such treatment significantly suppresses tumorigenicity of HEY xenografts through downregulation of N-cadherin and *Snail*, *Slug*, *Twist* transcription factors, as well as upregulation of E-cadherin expression [140] (Table 2).

### 4.11. Head and Neck Cancer

In gefitinib-resistant Hep-2 and KB squamous cell carcinoma of head and neck cells, SAHA reverted EMT by a time-dependent upregulation of E-cadherin and β-catenin and downregulation of vimentin. Moreover, there has been observed a reduction of the spindle-like morphology, characteristic for mesenchymal cells, with the acquisition of epithelial morphology, in both Hep-2 and KB cells after SAHA treatment [141]. VPA also induced a reversal of the mesenchymal phenotype caused by TGF-β1 or irradiation in TE9 cancer cells, resulting in an increase of cell migration and invasion. TE9 cancer cells pre-treated with VPA exhibited less inhibition of E-cadherin expression and no increase of vimentin expression as compared with untreated cells stimulated by TGF-β1 or irradiation. VPA inhibited the phosphorylation of *SMAD2* and *SMAD3* and downregulated *TWIST*, *SNAIL*, and *SLUG* transcription factors expression which previously were increased by TGF-β1 or irradiation stimulation [142] (Table 2).

### 4.12. Malignant Glioma

Panobinostat (LBH589) combined with temozolomide and irradiation stimulation significantly decreased vasculogenic mimicry (VM) formation, migration and invasion as well as increased E-cadherin expression in U251 glioma cells compared with temozolomide and irradiation stimulation without LBH589 treatment [143] (Table 2).

## 5. Discussion

Virtually all HDIs block multiple HDACs, which in turn have each multiple protein targets and thus the resulting gene expression changes are not the direct targets of HDIs but rather the downstream effects lack of different HDACs activities. This is probably why in many cases both epithelial as well as mesenchymal markers increase in expression. Since each HDI has different HDAC targets, the downstream results might be substantially different depending on the expression of HDACs, their cellular roles, as well as the expression of other co-factors or orthogonal signaling pathways.

Since in tumor cells often epigenetically inactivate non-beneficial genes, while activate those that provide them with evolutionary advantages, changes in gene expression can result in a beneficial effect due to the re-expression of tumor suppressor [147] and the transcriptional silencing of pro-oncogenes [148,149]. Moreover, changes in acetylation of non-histone proteins might also provide beneficial in many cases, such as the reduction pro-oncogenic or pro-survival signaling pathways.

The search for inhibitors of single HDACs might provide us with a clearer picture of why there are contradictory results depending of the cell type. Also, there is need to understand the effects of HDIs in different cancer cells in light of their HDAC expression patterns and genomic, as well as epigenomic, landscapes.

While there is evidence that de-novo gene expression, in particular of epithelial-like genes, is a beneficial result for the treatment, this very effect of randomly re-opening chromatin might have two other effects: (1) turning on oncogenes or transposons, the latter related to the next point; and (2) genomic restructuring which might have effects in genomic stability. Genomic stability should be carefully considered in future studies using HDIs as long-term treatment with these drugs might result in genomic instability and acquisition of mutation in re-opened chromatin. There is evidence that such might occur, promoting more aggressive tumors [150]. This also prompts that the length of HDIs treatments should be carefully analysed.

## 6. Conclusions

HDACs play a pivotal role in the progression of cancers by reversible modulation of acetylation status of histone and non-histone proteins. However, the exact function of HDACs as a central mediator of tumorigenic capacity still remains unclear. There are abundant pre-clinical and clinical studies examining the effects of HDIs alone or in combination with other anti-cancer agents. The impact of HDIs on EMT differ greatly in various types of cancers (Table 3). VPA stimulates EMT in hepatocellular [120], breast [112] and colorectal cancers [128] while the same active agent inhibits this process in lung [117], prostate [11] and head and neck carcinomas [142]. Moreover, the impact of VPA on EMT in renal cancer is unclear. After VPA treatment expression both epithelial and mesenchymal markers decreased, moreover migration of renal cells was inhibited [130]. In lung [33,114,115] and breast cancers [14] most HDIs inhibited EMT with the exception of SAHA, which stimulated this process in both mentioned above types of cancer. LBH589 suppresses EMT in breast cancer [14] and hepatocellular carcinoma [122]. However, in hepatocellular carcinoma most of HDIs (VPA, SAHA, TSA, MS-275 [120], NaB [121]) stimulated EMT, excluding LBH589 [122] and RAS2410 [123] which inhibited this process. In head and neck cancer [141,142] as well as prostate cancer all analyzed HDIs [11,13] inhibited EMT. Existing data about the effects of HDIs on EMT are conflicting. Therefore, there is an urgent need to comprehensively study the mechanisms of action and role of HDIs on the EMT/MET processes in different cancer types at different stages of carcinogenesis. Side effects of application of these compounds also should be noted. More studies are needed to establish the best strategy to incorporate these agents into the therapy of patients with cancers, minimizing toxicity and maximizing clinical benefits. Clarification and validation of the detailed mechanisms of HDIs action will provide a bright future for the use of HDIs as one of the important tools in the fight against cancers.

## Figures and Tables

**Figure 1 cancers-11-00148-f001:**
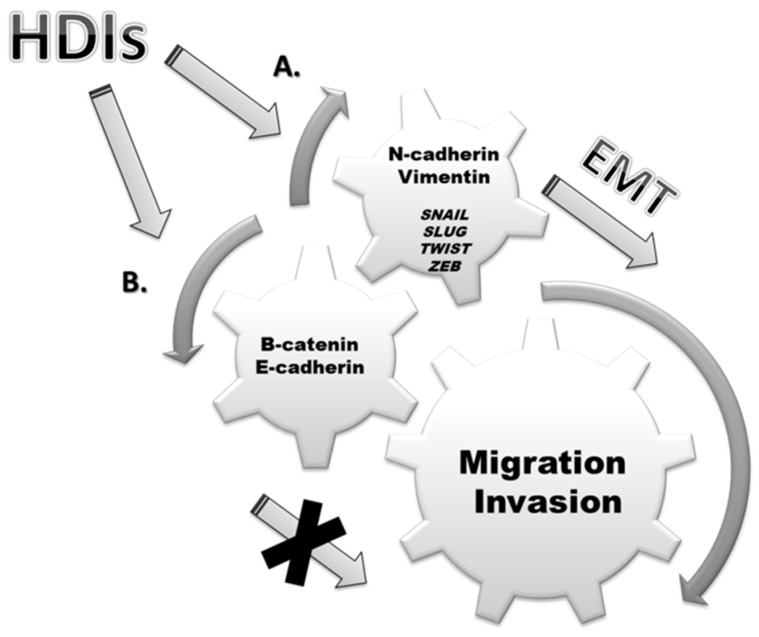
Histone deacetylase inhibitors (HDIs) modulate expression of epithelial-mesenchymal transition (EMT) markers as well as stimulate or inhibit migration and invasion of cancer cells. (**A**) HDIs induce EMT by increasing migration and invasion of cancer cells by upregulation of mesenchymal markers (N-cadherin, vimentin) and EMT-related transcription factors (*SNAIL*, *SLUG*, *TWIST*, *ZEB*). (**B**) HDIs upregulate expression of epithelial markers (E-cadherin, β-catenin) and consequently inhibit EMT, migration and invasion of cancer cells.

**Figure 2 cancers-11-00148-f002:**
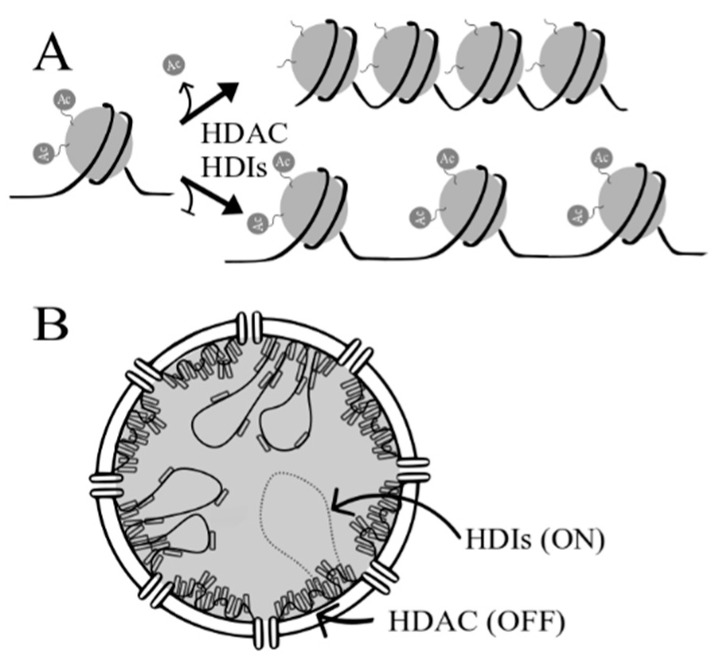
(**A**) Effect of histone deacetylase inhibitors (HDIs) on chromatin remodeling. Acetylation (Ac) of histones results in changes in chromatin conformation, where non-acetylated histones form heterochromatin (close chromatin) while acetylated histones result in relaxed chromatin—allowing DNA-binding by transcription factors. (**B**) Chromosomal landscape in the nucleus in the presence and absence of HDIs. Closed chromatin is near the nuclear envelope, while relaxed chromatin, where transcription is possible, is found in the middle of nucleus.

**Figure 3 cancers-11-00148-f003:**
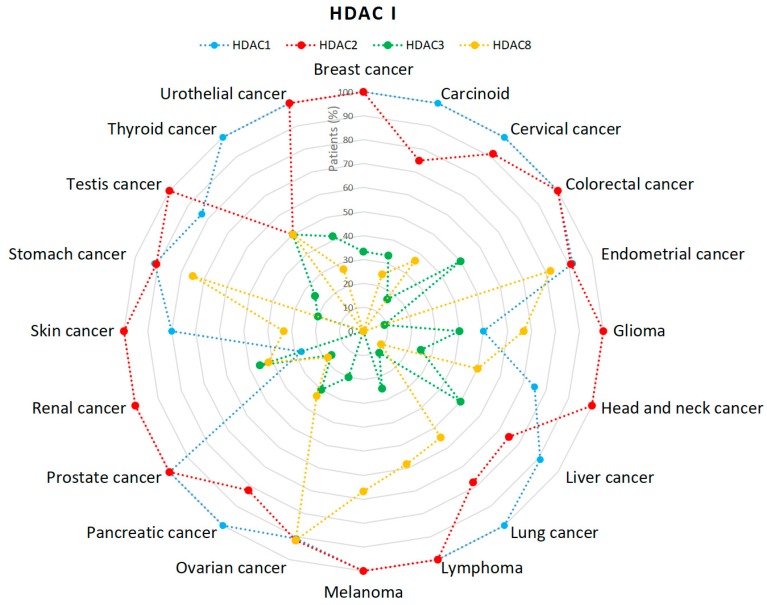
Percentage (%) of patients with high or medium HDACs class I expression levels in different types of cancer [43,44,45,46].

**Figure 4 cancers-11-00148-f004:**
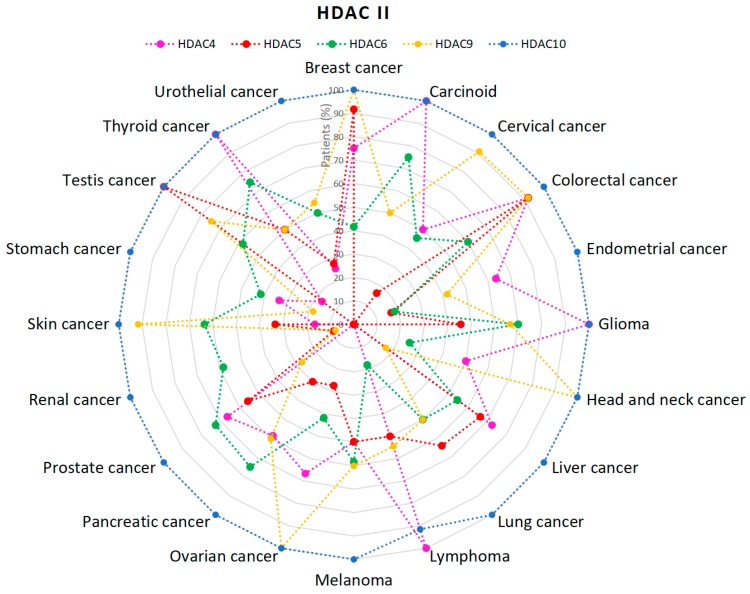
Percentage (%) of patients with high or medium HDACs class II expression levels in different types of cancer. HDAC7 was not analyzed [47,48,49,50,51,52].

**Figure 5 cancers-11-00148-f005:**
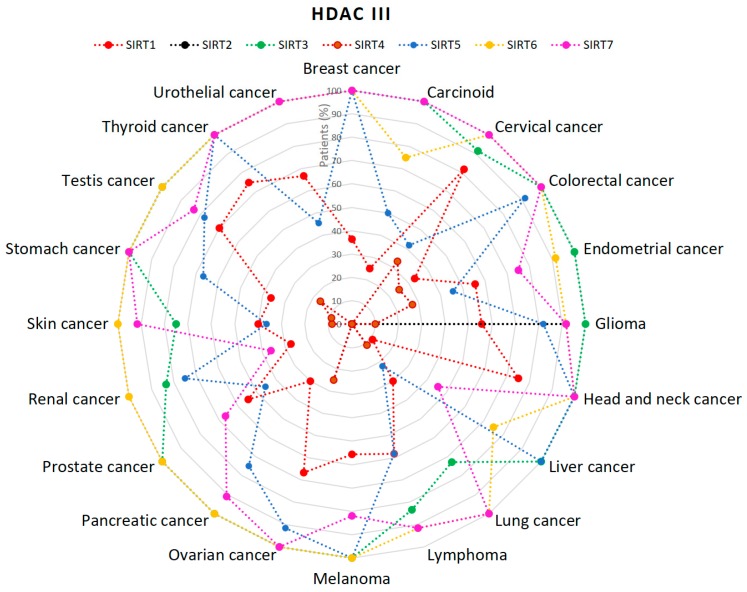
Percentage (%) of patients with high or medium HDACs class III expression levels in different types of cancer. There is no available data regarding the expression of HDAC11 (HDAC IV) [53,54,55,56,57,58,59,60].

**Figure 6 cancers-11-00148-f006:**
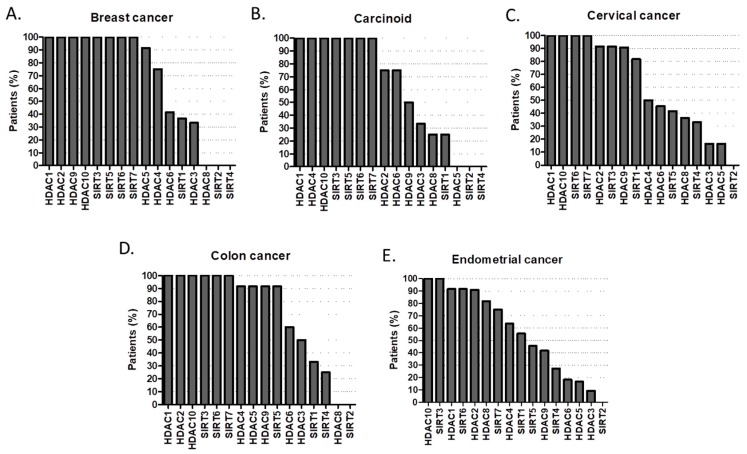
Percentage (%) of tumors with high or medium HDAC protein expression levels in (**A**) breast cancer, (**B**) carcinoid, (**C**) cervical cancer, (**D**) colon cancer, (**E**) endometrial cancer [43,44,45,46,47,48,49,50,51,52,53,54,55,56,57,58,59,60].

**Figure 7 cancers-11-00148-f007:**
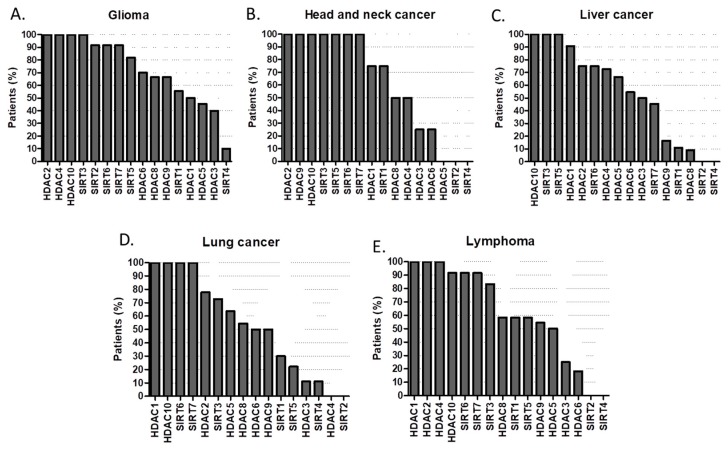
Percentage (%) of tumors with high or medium HDAC protein expression levels in (**A**) glioma, (**B**) head and neck cancer, (**C**) liver cancer, (**D**) lung cancer, (**E**) lymphoma [43,44,45,46,47,48,49,50,51,52,53,54,55,56,57,58,59,60].

**Figure 8 cancers-11-00148-f008:**
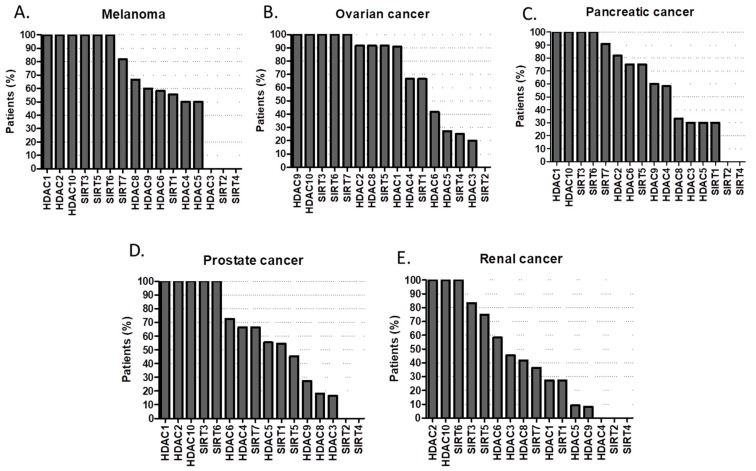
Percentage (%) of tumors with high or medium HDAC protein expression levels in (**A**) melanoma, (**B**) ovarian cancer, (**C**) pancreatic cancer, (**D**) prostate cancer, (**E**) renal cancer [43,44,45,46,47,48,49,50,51,52,53,54,55,56,57,58,59,60].

**Figure 9 cancers-11-00148-f009:**
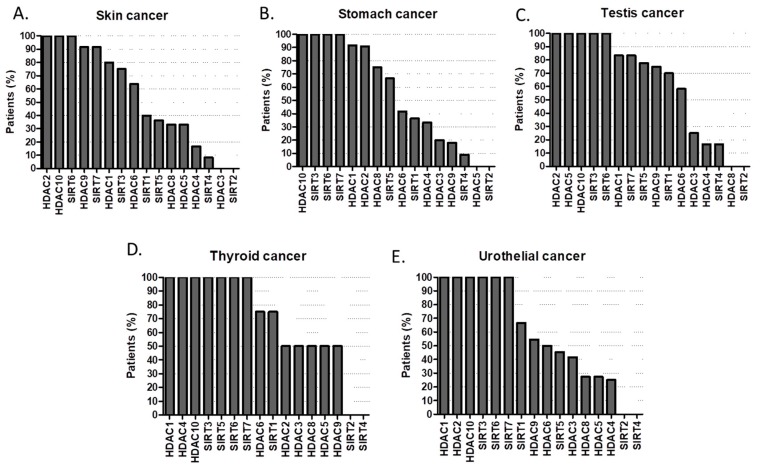
Percentage (%) of tumors with high or medium HDAC protein expression levels in (**A**) skin cancer, (**B**) stomach cancer, (**C**) testis cancer, (**D**) thyroid cancer, (**E**) urothelial cancer [43,44,45,46,47,48,49,50,51,52,53,54,55,56,57,58,59,60].

**Figure 10 cancers-11-00148-f010:**
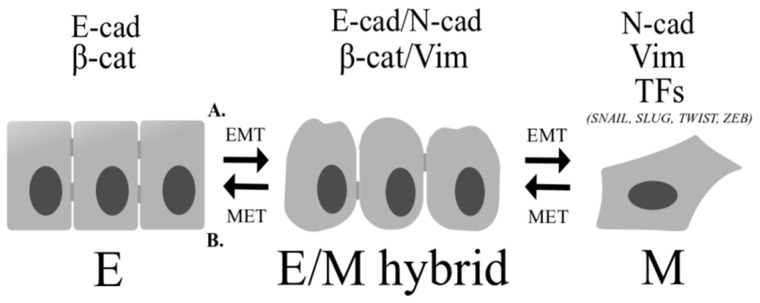
Phenotypical transformation of cells during the epithelial–mesenchymal transition (EMT) and mesenchymal-epithelial transition (MET) processes. (**A**) During EMT epithelial cells lose their polarized organization and acquire migratory and invasive capabilities by increase in mesenchymal markers (N-cadherin, vimentin) and EMT-related transcription factors (TFs) (*SNAIL*, *SLUG*, *TWIST*, *ZEB*). (**B**) During MET cells re-acquire epithelial properties. Epithelial cells are connected by intercellular junctions and they exhibit apical-basal polarization. The intermediate stage between fully-epithelial and fully-mesenchymal states has been described as E/M hybrid state. Cancer cells with E/M hybrid phenotype have cell-cell adhesion properties as well as migration abilities, simultaneously. E: epithelial; E/M hybrid: epithelial/mesenchymal hybrid; M: mesenchymal; E-cad: E-cadherin; β-cat: β-catenin; N-cad: N-cadherin; Vim: vimentin; TFs: transcription factors; *SNAIL*, *SLUG*, *TWIST*, *ZEB*: mesenchymal transcription factors.

**Table 1 cancers-11-00148-t001:** Histone targets of histone deacetylase inhibitors (HDIs).

Class of HDI	HDI	HDAC Targets	Ref.
Short chain fatty acid	Phenylbutyrate (PBA)	Pan-inhibitor	[84]
	Sodium butyrate (NaB)	I, IIa	[85]
	Butyrate	I, IIa	[83]
	Valproic acid	I, IIa	[86]
Hydroxamic acid–derived compounds	Vorinostat (SAHA)	Pan-inhibitor	[87]
	Belinostat (PXD-101)	Pan-inhibitor	[88]
	Resminostat (4SC-201)	Pan-inhibitor	[83]
	Panobinostat (LBH589)	I, II	[83]
	Trochostatin A (TSA)	I, II	[24]
Benzamides	Entinostat (MS-275)	I	[89]
	Mocetinostat (MGCD103)	I	[90]
	Domatinostat (4SC-202)	I	[80]
Cyclic peptides	Romidepsin (FK228)	I	[91]
	Apicidin (CAS183506-66-3)	I	[83]

**Table 2 cancers-11-00148-t002:** Influence of histone deacetylase inhibitors (HDIs) on epithelial-mesenchymal transition (EMT) markers, transcription factors, morphology, migration and invasion of cancer cells in vitro and in vivo.

Type of Cancer	HDI (Individually or in Combination)	Experimental Model	Type of Treatment	E-cadherin	B-catenin	N-cadherin	Vimentin	Transcription Factors	Changes in Morphology	Migration and Invasion	Ref.
Lung cancer	SAHA	A549 cells in vitro	cells treated with SAHA vs. untreated cells	↓	→	N/A	↑	↑*SLUG*	from cobblestone to mesenchymal spindle-like	↑migration	[118]
Lung cancer	TSA	A549 cells in vitro	irradiated cells treated with TSA vs. irradiated cells	↑	↑	↓	↓	↓*SNAIL*, *ZEB*	reduction of mesenchymal-like phenotype	↓migration	[114] [115]
Lung cancer	TSA + silibinin	H1299 cells in vitro	cells treated with TSA and silibinin vs. cells treated with silibinin	↑	N/A	N/A	N/A	↓*ZEB1*	N/A	↓migration and invasion	[116]
Lung cancer	VPA	A549 cells in vitro	cells treated with VPA vs. untreated cells	↑	N/A	N/A	N/A	N/A	reduction of spindle-like morphology	N/A	[117]
Hepatocellular carcinoma	TSA	HepG2 cells, Huh7 cells in vitro	cells treated with TSA vs. untreated cells	↓	N/A	↑	↑	↑*SNAIL*, *TWIST*	N/A	↑migration and invasion	[120]
Hepatocellular carcinoma	VPA	HepG2 cells, Huh7 cells in vitro	cells treated with VPA vs. untreated cells	↓	N/A	↑	↑	↑*SNAIL*, *TWIST*	N/A	↑migration and invasion	[120]
Hepatocellular carcinoma	SAHA	HepG2 cells in vitro	cells treated with SAHA vs. untreated cells	↓	N/A	↑	↑	↑*SNAIL*, *TWIST*	N/A	↑migration and invasion	[120]
Hepatocellular carcinoma	MS-275	HepG2 cells in vitro	cells treated with MS-275 vs. untreated cells	↓	N/A	↑	↑	↑*SNAIL*, *TWIST*	N/A	↑migration and invasion	[120]
Hepatocellular carcinoma	SAHA	HepG2 cells, QGY-7703 cells in vitro; mouse in vivo	cells treated with SAHA vs. untreated cells	N/A	N/A	↑	↑	↑*SNAIL* through *SMAD2/3* phosphorylation	changes of phenotype were detected	↑invasion	[121]
Hepatocellular carcinoma	NaB	HepG2 cells/QGY-7703 cells in vitro; mouse in vivo	cells treated with NaB vs. untreated cells	N/A	N/A	N/A	↑	↑*SNAIL* through *SMAD2/3* phosphorylation	N/A	↑invasion	[121]
Hepatocellular carcinoma	LBH589	HepG2 cells in vitro	cells treated with LBH589 vs. untreated cells	↑	N/A	↓	↓	↓*TWIST1*	N/A	↓invasion	[122]
Hepatocellular carcinoma	RAS2410	Hep3B, HLE, HLF cells in vitro	cells treated with RAS2410 vs. untreated cells	↑	N/A	↓	↓	→*TWIST*, *SNAI1*	N/A	↓migration and invasion	[123]
Cholangiocarcinoma	VPA	HuCC-T1 cells in vitro	cells treated with VPA vs. untreated cells	→	N/A	N/A	→	N/A	no changes	↓migration and invasion	[124]
Cholangiocarcinoma	TSA	HuCC-T1 cells in vitro	cells treated with TSA vs. untreated cells	↑	N/A	N/A	↑	N/A	no changes	↓migration and invasion	[124]
Cholangiocarcinoma	VPA + gemcitabine	HuCC-T1 cells in vitro	cells treated with VPA and gemcitabine vs. cells treated gemcitabine	↑	N/A	N/A	↑	N/A	from spindle to rectangular caused by gemcitabine	↓migration and invasion	[124]
Cholangiocarcinoma	TSA + gemcitabine	HuCC-T1 cells in vitro	cells treated with TSA and gemcitabine vs. cells treated gemcitabine	↑	N/A	N/A	↑	N/A	from spindle to rectangular caused by gemcitabine	↓migration and invasion	[124]
Pancreatic cancer	4SC-202	Panc1 cells L3.6 cells in vitro	TGF-β1 pretreated Panc1 cells treated with 4SC-202 vs. untreated cells in vitro; mice with implanted L3.6 cells in vivo	↓	N/A	↑	↓	↓*ZEB1*, *SNAIL1*	N/A	N/A	[125]
Pancreatic cancer	BSI	Panc1 cells in vitro	Panc1 cells treated with BSI vs. untreated cells in vitro	↑	N/A	↓	N/A	↓*SNAIL*	tumor spheres formation is unchanged but their size is significantly decreased	↓migration and invasion	[126]
Pancreatic cancer	MGCD103 + gemcitabine	Panc1 cells, hPaca-1 derived tumor cells in vitro	Panc1 cells, hPaca-1 derived tumor cells treated with MGCD103 and gemcitabine vs. gemcitabine treated cells in vitro	↑	N/A	N/A	N/A	↓*ZEB1*	N/A	N/A	[127]
Pancreatic cancer	SAHA	Pancreatic CSCs	pancreatic CSCs treated with SAHA vs. untreated cells in vitro	↑	N/A	↓	N/A	↓*ZEB*, *SNAIL*, *SLUG*	N/A	↓invasion	[12]
Colorectal cancer	TSA	SW480 cells in vitro	cells treated with TSA vs. untreated cells	↑	N/A	N/A	↓	↓*SLUG*	N/A	↓migration and invasion	[121]
Colorectal cancer	VPA	SW480 cells in vitro	cells treated with VPA vs. untreated cells	↓	N/A	↑	↑	↑*SNAIL*	N/A	↑migration and invasion	[128]
Colorectal cancer	VPA	HCT116 cells in vitro	cells treated with VPA vs. untreated cells	↓	N/A	↑	↑	↑*SNAIL*	N/A	↑migration and invasion	[128]
Colorectal cancer	Compound 11	HCT116 cells in vitro	cells treated with compound 11 vs. untreated cells	↑	↓	↓	↓	N/A	N/A	↓migration	[129]
Colorectal cancer	Compound 11	HT29 cells in vitro	cells treated with compound 11 vs. untreated cells	N/A	N/A	N/A	N/A	N/A	N/A	↓migration	[129]
Colorectal cancer	Compound 11	HCT116 xenograft model in vivo	mice treated compound 11 vs. untreated mice	↑	N/A	↓	↓	N/A	N/A	↓migration	[129]
Colorectal cancer	TSA	HT29, SW480, DLD1, HTC116 cells in vitro	cells treated with TSA vs. untreated cells	↓	N/A	N/A	↑	N/A	altered to spindle like morphology	→migration, ↑invasion only in DLD1 cells	[15]
Colorectal cancer	VPA	HT29, SW480, DLD1, HTC116 cells in vitro	cells treated with VPA vs. untreated cells	↓	N/A	N/A	↑	N/A	altered to spindle like morphology	→migration, ↑invasion in DLD1 and SW480 cells	[15]
Colorectal cancer	TGF-β1	HT29, SW480, DLD1, HTC116 cells in vitro	cells treated with TGF-β1 vs. untreated cells	↓	N/A	N/A	↑	N/A	altered to spindle like morphology	↑invasion only in DLD1 cells	[15]
Colorectal cancer	TSA+ TGF-β1	HT29, SW480, DLD1, HTC116 cells in vitro	cells treated with TSA and TGF-β1 vs. untreated cells	↓	N/A	N/A	↑	N/A	altered to spindle like morphology	HT29 N/A, SW480 ↑migration, LDL1 →invasion, HTC116 N/A	[15]
Colorectal cancer	VPA + TGF-β1	HT29, SW480, DLD1, HTC116 cells in vitro	cells treated with VPA and TGF-β1 vs. untreated cells	↓	N/A	N/A	↑	N/A	altered to spindle like morphology	HT29 N/A, SW480 ↑migration, LDL1 ↑migration, →invasion, HTC116 N/A	[15]
Renal cancer	VPA	Renca cells in vitro, mice in vivo	cells treated with VPA vs. untreated cells	↓	↓	N/A	↓	↑*TWIST1*, ↓*TWIST2* →*SNAIL1*, *SNAIL2*	interspace between cells after HDIs treatment	↓migration	[130]
Renal cancer	MS-275	Renca cells in vitro, mice in vivo	cells treated with MS-275 vs. untreated cells	↓	↓	N/A	N/A	N/A	interspace between cells after HDIs treatment	↓migration	[130]
Renal cancer	TSA	HK2 cells in vitro	TGF-β1-pretreated HK2 cells treated with TSA vs. TGF-β1-treated HK2 cells	↑	N/A	→	N/A	N/A	N/A	N/A	[131]
Renal cancer	TSA	RPTEC cells in vitro	TGF-β1-pretreated RPTEC cells treated with TSA vs. untreated RPTEC cells	↑	N/A	N/A	N/A	→*SMAD2*, *SMAD3*	from cuboidal to elongated form	N/A	[131]
Urothelial cancer	CDDP+SAHA	RT-112 and T-24 cells in cell culture or implanted on the chicken chorioallantoic membrane (CAM)	cells implanted on the CAM treated with CDDP + SAHA vs. cells treated with CDDP	N/A	N/A	N/A	N/A	N/A	CAM tumor reduction		[132]
Urothelial cancer	CDDP+Romidepsin	RT-112 and T-24 cells in cell culture or implanted on the chicken chorioallantoic membrane (CAM)	cells implanted on the CAM treated CDDP+Romidepsin vs. cells treated with CDDP	N/A	N/A	N/A	N/A	N/A	CAM tumor reduction		[132]
Prostate cancer	AR-42	Ace-1 cells in vitro	cells treated with AR-42 vs. untreated cells	↓	→	↓	→	↓*TWIST*, *MYOF*, ↑*SNAIL*, *SLUG*, *PTEN*, *FAK*, *ZEB1*	reduction of spindle like morphology	↓migration and invasion	[133]
Prostate cancer	AR-42	nude mice with implanted Ace-1 cells in vivo	mice with Ace-1 cells treated AR-42 vs. untreated mice	N/A	N/A	N/A	N/A	N/A	irregular shape of cell after AR42 treatment	↓reduction of bone metastasis	[133]
Prostate cancer	SAHA, TSA, RGFP966	LNCaP cells in vitro	cells treated with HDIs vs. untreated cells	N/A	N/A	N/A	↑SAHA, TSA; →RGFP966	↓*NKX1*, *FOXA1*; ↑*SLUG*, *ZEB1* (SAHA, TSA), →*SLUG*, *ZEB1* (RGFP966)	N/A	↑ migration (SAHA), N/A (TSA), →migration (RGFP99)	[134]
Prostate cancer	TSA	PC3 cells in vitro	cells treated with TSA vs. untreated cells	↑	N/A	N/A	↓	↓*SLUG*	N/A	↓migration and invasion	[13]
Prostate cancer	VPA	PC3 cells in vitro	cells treated with VPA vs. untreated cells	↑	N/A	N/A	N/A	N/A	N/A	↓migration	[11]
Breast cancer	SAHA	MzChA-1 and TFK-1 cells in vitro	cells treated with SAHA pretreated with TGF-β1 vs. cells treated with TGF-β1	↑	N/A	↓	↓	inhibition of *p-SMAD2*, *p-SMAD3* and *SMAD4* nuclear translocation induced by TGF-β1	reduction of changes from valvate-like- to spindle-like shapes caused by TGF-β1	N/A	[135]
Breast cancer	SAHA	MDA-MB-231 and BT-549 cells in vitro	cells treated with SAHA vs. untreated cells	↓	N/A	↑	↑	→*SNAIL*, *SLUG*, *TWIST* and *ZEB* expression and translocation	N/A	↑migration	[136]
Breast cancer	SAHA, VPA	MDA-MB-231 and SUM159 cells in vitro	ed with VPA or SAHA vs. untreated cells	not detected	N/A	↑	↑	↓*FOXC3*, *ZEB1* ↑*SNAIL2*, *TWIST1*	↑sphere formation	↑migration	[137]
Breast cancer	LBH589	MDA-MB-231 and BT-549 cells in vitro	cell treated with LBH589 vs. untreated cells	↑	N/A	↓	↓	↓*ZEB1*, *ZEB2*	more epithelial phenotype	↓migration and invasion	[138]
Breast cancer	LBH589	MCF7 cells in vitro	cell treated with LBH589 vs. untreated cells	→	N/A	N/A	→	→*ZEB1*, *ZEB2*	more epithelial phenotype	↓migration and invasion	[138]
Breast cancer	MS-275	MDA-MB-231 and Hs578T cells in vitro	cells treated with MS-275 vs. untreated cells	↑	N/A	↓	↓	↓*SNAIL*, *TWIST*	more epithelial phenotype	↓migration	[14]
Breast cancer	MS-275	Balb c nude mice implanted with TRAIL resistant MDA-MB-468 cells in vivo	mice treated MS-275 vs. untreated mice	↑	N/A	N/A	↓	↓*ZEB1*, *SNAIL*, *SLUG*	N/A	N/A	[139]
Breast cancer	MS-275+TRAIL	Balb c nude mice implanted with TRAIL resistant MDA-MB-468 cells in vivo	mice treated MS-275+TRAIL vs. mice treated TRAIL only	↑	N/A	N/A	↓	↓*ZEB1*, *SNAIL*, *SLUG*	N/A	N/A	[139]
Ovarian cancer	TSA	SKOV3 cells in vitro	cells treated with TSA vs. untreated cells	↓	N/A	N/A	↓	N/A	N/A	↓migration	[140]
Ovarian cancer	TSA+cisplatin	SKOV3 cells in vitro	cells treated with TSA + cisplatin vs. untreated cells	↓	N/A	N/A	↓	N/A	N/A	↓migration	[140]
Ovarian cancer	TSA+cisplatin	Mice with HEY injected cells in vivo	mice treated with cisplatin followed by TSA vs. untreated mice	↑	N/A	N/A	↓	↓*SNAIL*, *SLUG*, *TWIST*	N/A	N/A	[140]
Head and neck cancer	SAHA	Hep-2 and KB cells in vitro	cells treated with SAHA vs. untreated cells	↑	↑	N/A	↓	N/A	reduction of the spindle like morphology	↓migration and invasion	[141]
Head and neck cancer	VPA	TE9 cells pretreated with TGF-β1 or irradiation in vitro	cells treated with VPA and TGF-β1 or irradiation before vs. cells treated with TGF-β1 or irradiation	↑	N/A	N/A	↓	↓*SMAD2* and *SMAD3* phosphorylation, ↓*TWIST*, *SNAIL*, *SLUG*	reduction of spindle like morphology caused by TGF-β1 or irradiation	↓migration and invasion	[142]
Malignant glioma	LBH589+irradiation	U251 cells in vitro	cells treated with LBH589+irradiation vs. untreated cells	↑	N/A	N/A	N/A	N/A	reduction of vasculogenic mimicry formation	↓migration and invasion	[143]

Abbreviations: ↑ increase, ↓ decrease, → no changes observed. SAHA-vorinostat, TSA-trichostatin A, VPA-valproic acid, MS-275 entinostat, NAB-sodium butyrate, LBH589-panobinostat, RAS2410-resminostat, 4SC-202-domatinostat, MGCD103-mocetinostat, compound 11-(E)-*N*-hydroxy-3-(1-(4-methoxyphenylsulfonyl)-1,2,3,4-tetrahydroquinolin-6-yl)acrylamide.

**Table 3 cancers-11-00148-t003:** The effect of histone deacetylase inhibitors (HDIs) on the epithelial-mesenchymal transition (EMT) process in tumors.

HDI	↑EMT	↓EMT	Unclear Mechanism
VPA	Hepatocellular carcinoma [120], breast [112], colorectal cancer [128]	Lung [117], prostate [11], head and neck cancer [142]	Renal cancer [130]
SAHA	Hepatocellular carcinoma [120], lung [118], breast cancer [136]	Pancreatic [12], head and neck cancer [141]	-
TSA	Hepatocellular carcinoma [120], colorectal cancer [128]	Lung [114,115], prostate cancer [13]	Cholangiocarcinoma [124], ovarian cancer [140]
MS-275	Hepatocellular carcinoma [120]	Breast cancer [14]	Renal cancer [130]
LBH589	-	Hepatocellular carcinoma [122], breast cancer [14]	-
RAS2410	-	Hepatocellular carcinoma [123]	-
4SC-202	-	-	Pancreatic cancer [125]
AR-42	-	-	Prostate cancer [133]
NaB	Hepatocellular carcinoma [121]	-	-
BSI	-	Pancreatic cancer [126]	-
Compound 11	-	Colorectal cancer [129]	-

Abbreviations: ↑ increase, ↓decrease, SAHA-vorinostat, TSA-trichostatin A, VPA-valproic acid, MS-275 entinostat, NAB-sodium butyrate, LBH589-panobinostat, RAS2410-resminostat, 4SC-202-domatinostat, MGCD103-mocetinostat, compound 11-(E)-*N*-hydroxy-3-(1-(4-methoxyphenylsulfonyl)-1,2,3,4-tetrahydroquinolin-6-yl) acrylamide.

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
