# Peer review of "Histone Deacetylase Inhibitors and Phenotypical Transformation of Cancer Cells"

_cancers, 2019, doi:10.3390/cancers11020148_

Round 1
Reviewer 1 Report
Wawruszak et al., reports the review about recent findings on the effect of histone deacetylase inhibitors (HDIs) on the epithelial-mesenchymal transition (EMT) process in human cancers. This review is well described and will provide some insights into the understanding of the roles of EMT targeted therapy of cancer by HDIs. However several points are should be modified for its publication in Cancers to make the manuscript useful to the readers. The specific points are as follow.
1. “2. Histone deacetylase inhibitors (HDIs)”. The authors should describe the characteristic features of each class of histone deacetylase (HDAC) including the difference of its functions (whither it targets histone or non-histone proteins).
2. In Figure 3, the position of “SNAIL, SLUG, TWIST, ZEB” should be reconsidered to make clear their roles as EMT transcription factors.
3. Table 1 should be more concise.
4. “4. EMT and cancers”. This part is itemized too much and the authors’ conclusion, “The impact of HDIs on EMT differ in various types of cancers”, is not explained clearly in this manuscript.
Author Response
All responses to the Reviewer can be found in the attachment.

Reviewer 2 Report
The review by Wawruszak attempted to target the effect of HDAC inhibitors in cancer cells. Though the authors tried to cover a number of cancer types with table/description, this article lacks a structure, particularly the way it introduces. This manuscript is noncomprehensive in several ways. Urothelial carcinoma is need to be added in the description passage of several cancers and table section.
In the earlier section, readers wish to learn the following: How many HDACs are there? Information about different classes… HDACs are enzymes, so what is its catalytic activity and what are the HDACs possess stron/less activity, and their target substrates. What are the other functions, Is deacetylation the only major function? And how that links to cancer. Why one wants to inhibit HDACs, here in different cancers? Because, in what cancer types different HDACs are upregulated (or downregulated)? Are all HDAC inhibitors inhibiting the function of different HDACs? What are their specificity? What are the known mechanistic understanding in different cancer types?- Authors may want to consider addressing them.
It is also unusual to have only 68 references in a 20-page review! Substantial attention is needed for citations, authors need to cite correctly, ONLY the relevant previous works and reviews, accordingly. Some examples:
1. Page 1, line 38, ref 1àVPA inhibits EMT. This will not help to the message said. Instead, seminal reviews on EMT are a good fit, not necessarily restricted to HDACi-induced EMT, as it is the first sentence of this review. Some examples: Kalluri and Weinberg 2009, Radisky DC 2005, ref 17 Lamouille et al , Davis et al 2014 trends in pharmacological sciences and others.
2. Line 44, similarly ref 3 and 4 do not support the context! These articles suggest the role of HDACis on EMT progression and not the anticancer effect. Relevant reviews and perhaps studies can be added instead of ref 3 and 4.
3. Page 2, line 45àCitation required for MET.
4. Just to be clear, VPA is one of the HDAC inhibitors. So authors cannot generalize by having one. Please find the relevant articles in the literature.
5. What do you mean by “nucleosome post-transcriptional modifications” in page 2, line 57? Ref 5 didn’t help me.
6. P2, Line 60, Phosphoryation is missing as taken from ref 7.
7. P2, L61 please cite, “HIstonce accetylation is one of the most…..”
8. P2, L65, please give some real examples of non-histone proteins that are affected by HDis
9. P3, L78, please insert citations, “..dysregulated in many types of cancers”
10. Need more reviews and papers to endorse the statement in p3, L80-84. Also the cited references need to be given in the appropriate context. Please do not generalize just because a drug works against one type of cancer.
11. This important information needs references, p3, L88-90. And the last sentence of the paragraph. Nothing is really unresolved? and more
Discussion and conclusion have to be improved.
Figure 1 would benefit from additional annotation to improve clarity. Hard to understand what the arrow is pointing from HDI to the first circle…anything EMT/MET and migration/invasion? 1
Minor comments
1. Romidepsin is an antibiotic, please metion in the class of HDIs in page 3
2. I recommend authors to go for professional English proofreding
Abstract and elsewhere-over usage of semicolons (;) in place of commas (,)
P2, L63, Typo, chromatin “confirmation”
P3, L7—“genes expression”
Please rewrite-p3, L75-77
Author Response

(The authors gave the same response as above.)

Round 2
Reviewer 1 Report
The authors adequately revised the manuscript according to the reviewers' comments.
Author Response
Dear Reviewer 1,
thank you very much for your valuable comments and for the time which you spent reviewing our manuscript.
With kind regards,
Anna Wawruszak
Reviewer 2 Report
I think authors have done a good revision and presented a much improved MS. Again, only one particular article was discussed in urothelial carcinoma (UC) description. Please cite and add briefly discuss the following important articles in the UC and other relevant section. To assist authors and simplify this passage I have looked and suggested them from one group only (as in ref. 128). Perhaps this comment stands true other cancer types discussed with ONE reference.
Kaletsch et al., 2018 Clin Epigenetics
Hölscher et al., 2018 Clin Epigenetics
Niegisch et al., 2013 Urol Oncol
Pinkerneil et al., 2016 mol. Can. Ther. and second paper in Target Oncol.
Lehmann et al., 2014 J Exp Clin Can Res
2. p3 line71--PTM--> typo still need to be corrected, histone post-transcriptional modification- do you mean methylation or acetylation of histone coding RNAs?
Author Response
Dear Reviewer 2,
we would like to thank you for pointing out those publications which we missed during our search. We have added them to the manuscript in the appropriate context.
We also explained the term “PTMs”. We changed “post-transcriptional” modifications to “post-translational” (line 71). Term “histone PTMs” we explained in the next sentence “Histone PTMs include methylation, phosphorylation, acetylation, sumoylation, ubiquitination and ADP-ribosylation”.
All the corrections and changes have been marked in green.
We would like to thank you once more for your valuable comments and corrections.
With kind regards,
Anna Wawruszak